# A Longitudinal Multi-Method Inquiry of Educational Workers’ Use of Interventions for Positive Mental Wellbeing

**DOI:** 10.3390/healthcare12222200

**Published:** 2024-11-05

**Authors:** Astrid Helene Kendrick, Mawuli Kofi Tay, Lisa Everitt, Rachel Pagaling, Shelly Russell-Mayhew

**Affiliations:** 1Werklund School of Education, University of Calgary, Calgary, AB T2N 1N4, Canada; 2Alberta Teachers Association, Edmonton, AB T5N 2R1, Canada

**Keywords:** compassion fatigue, burnout, personal and professional interventions for treatment, positive mental health, outdoors

## Abstract

Background and Objectives: Compassion fatigue and burnout are two distinct forms of mental health distress faced by educational workers. Researchers have shown a high level of both phenomena across the field of education; however, a better understanding of what educational workers already do for positive mental and emotional health is needed. Methods: This research study examined three years of data, collected via survey, inquiring into the various interventions, namely supports and resources, that educational workers use to support positive mental health. Results: Quantitative data analysis via descriptive and inferential statistics revealed that educational workers relied heavily on their personal support network followed distantly by medical professionals and other interventions, revealing a gap that needs to be addressed by employers. Qualitative thematic analysis revealed a trend towards increased use of environmental interventions to promote positive mental wellbeing. Conclusions: The data analysis suggested areas of focus required to ensure workplace wellbeing, and that programs too focused on individual or self-directed interventions would not be well received by educators. Suggestions for other interventions that might be helpful for leaders and policy-makers are provided.

## 1. Introduction

Improving workplace wellbeing through integrating positive mental health practices has become a growing concern for educational leaders. Two known psychological hazards for educational workers are burnout and secondary traumatic stress or secondary traumatic stress disorder, known as compassion stress and compassion fatigue, respectively. Traumatic events, such as the COVID-19 pandemic, widespread forest fires, and flooding, have resulted in high levels of both compassion fatigue and burnout in Alberta’s educational workers as they increasingly face increased workloads and decreased classroom supports and resources [1]. Understanding the treatments, supports, and resources used by educational workers when they become distressed is an important starting point for developing and implementing effective workplace wellbeing programs [2].

However, workplace wellbeing programs can be grounded in varied definitions of wellness, wellbeing, and health [3]. Wellness has been defined as “a multidimensional state of being describing the existence of positive health in an individual as exemplified by quality of life and sense of well-being” [2]. The term has been used since the time of Socrates and gained in popularity through the work of Florence Nightengale [4]. Several theoretical models exist, and many of them place self-responsibility as central to developing wellness [5,6]. In these models, wellness is seen as a positive outcome of health, and an individual attains this state through balancing multiple dimensions such as nutrition, physical activity, stress, context, and spirituality.

More recently, the shift towards understanding collective wellbeing has changed the narrative from individual responsibility to a greater focus on the supports and resources available from the wider system and organization to improve overall health and wellbeing. Wellbeing, a derivative of “be well”, refers generally to a person being in a positive state. This research study frames workplace wellbeing as a state of being specific to one’s job and work role. Functioning well at work is a component of overall health underscoring that educational institutions and systems influence individual wellbeing and individuals influence the overall culture and the social and physical environments [3,7].

Given the reciprocal relationship between individual and system wellbeing, understanding the supports and resources used by educational workers when they experience distress might provide insights to employers and school leaders interested in improving workplace wellbeing. In this study, the categories of supports and resources are labelled broadly as interventions that could be taken on the level of individual/self-directed, school/site-based, organizational/system, professional/expert, and educational/field.

Data collected at three points in the years 2020 to 2023 were analyzed to explore the following question: Which forms of intervention were most and least commonly used by educational workers to be well at work and what trends are apparent in the data?

### 1.1. History of Workplace Wellbeing

In Canada, workplace wellbeing is a relatively new phenomena that has evolved alongside the establishment of universal health care and competitive labour markets [8]. The first universal health care coverage was founded by Premier Tommy Douglas in 1947, initially covering hospital care; however, by 1965, under Prime Minister Lester Pearson, the federal government implemented a broader model with provinces meeting the standards of comprehensiveness, portability, universality, and public administration ([9], p. 1718). At that time, most health plans met only basic needs aligned with expanding social security benefits, largely intended to attract employees [8].

The traditional model for health care up to the end of the 1960s was focused on curing, not preventing disease [10]. However, by 1974, the seminal Lalonde report was released that expanded the traditional model of health care leading to “the reconceptualization of public health policy” ([10], p. 534). This Health Field Concept was a new lens for thinking about public health care policy, comprised of four key elements: human biology, environment, lifestyle, and health care organization. Lalonde (1974) wrote that “one of the evident consequences of the Health Field Concept has been to raise HUMAN BIOLOGY, ENVIRONMENT, and LIFESTYLE to a level of categorical importance equal to that of HEALTH CARE ORGANIZATION” (p. 31). Rather than focusing solely on hospitals and doctors to cure illness, other preventative factors such as research on human biology, promotion of clean safe environments, and evaluation of the impacts of individual lifestyle choices became more important in improving workplace wellbeing. Kirk, Tomm-Bonde, and Schreiber [11] found that the Health Field Concept pillar used most often in health promotion was individual responsibility, not societal or structural changes.

A major shift in health promotion philosophy occurred in 1986 through the *Ottawa Charter for Health Promotion*. This document was produced by the World Health Organization (1986) [12].

Changing patterns of life, work, and leisure have a significant impact on health. Work and leisure should be a source of health for people. The way society organizes work should help create a healthy society. Health promotion generates living and working conditions that are safe, stimulating, satisfying, and enjoyable (p. 2).

The Ottawa Charter highlighted that peace, shelter, education, and food were critical components for peoples’ positive wellbeing. However, as the Ottawa Charter was aspirational in nature without specific policy suggestions, it was not adopted widely until the Comprehensive School Health movement in the 2010s [11].

These pivotal policy documents that underpin the wellbeing of Canadians, the Ottawa Charter, and Lalonde Report [13,14] focus primarily on physical wellbeing, sidelining mental and emotional wellbeing. This focus was then mirrored in employee benefit plans that tend to place the onus for mental emotional health on individual employees rather than collective or organizational responsibility. As noted by Morneau Sobeco [8], many short- and long-term disability plans have exclusions for disabilities arising from attempted suicide or self-inflicted injuries (p. 370). 

A significant shift for health promotion at work in Canada came in 2013 with the introduction of the *National Standard of Canada for Psychological Health and Safety in the Workplace* that explicitly directed managers and employers to minimize psychological harm at the workplace. This policy statement placed the harms of compassion fatigue and burnout as an organizational responsibility alerting employers to:

“[Avoid] reasonably foreseeable harm—through the establishment of a psychologically safe system of work—as the base level of care to which employers should aspire. The standard advises that mental injury is a type of harm that can and should be prevented by making every reasonable effort to ensure fairness, respectfulness, and consideration as cardinal values driving everyday interactions and practices in the workplace”.([11], p. 20)

Recent legislative changes that require both physical and mental safety in the workplace shift the onus for wellbeing from strictly that of individuals to all employees and employers. School administrators are now accountable for developing a safe and caring workplace for numerous adults, so understanding the interventions available to them has increased in importance of late. 

In Alberta, Canada, the Health and Physical Education curriculum has long included Comprehensive School Health as a framework for promoting healthier schools [15]. More recently, a new emphasis on improving educator wellbeing has emerged as a critical aspect of ensuring that children and youth are well-supported, as mentally and emotionally well adults are critical for ensuring safe and caring classrooms for Kindergarten to Grade 12 children and youth [12]. However, Alberta teachers have very high levels of both compassion fatigue and burnout, requiring specific intervention to ensure they have the internal capacity to care for their students [1,14].

### 1.2. Theoretical Framework: Levels of Intervention, from Individual to Collective

Training for caregiving professions has recently stressed the importance of an ecological approach to workplace wellbeing. Because wellbeing is complex and involves multiple connected actors, a complex adaptive systems theoretical orientation [16,17] was used to frame this research study. Within complex adaptive systems thinking, educational systems are not characterized as static or predictable entities, rather they change, innovate, and stagnate due to the complex interplay of the internal and external people within the system itself [15,16].

Initially, complexity thinking emerged from Brofenbrenner’s Ecological Model (1975) that suggested human development could not be viewed in isolation from the individual’s environment. Addressing the micro (individual), meso (school/workplace level), and macro (system/provincial) level influences [18,19,20] is an important part of developing, implementing, and studying workplace wellbeing. 

At the micro level, individual people work within different work roles, such as teacher, educational assistant, support staff, principal, or school district leader. These individuals often adhere to the belief that their work is important both on students currently in schools and on the future of society [18,20]; therefore, they willingly take on the task of creating safe and caring classrooms and schools for children and youth. 

In an effort to create a caring classroom environment, educational workers may seek to hide emotions such as frustration, anger, or sadness from their colleagues, staff, or students to align with organizational feeling rules [18]. In educational settings, organizational feeling rules are crafted both explicitly through documents such as the Teaching Quality Standard [21] and implicitly through staff induction and mentorship [22]. This emotional repression to align with organizational feeling rules is a form of superficial acting [23,24] that has been linked to increased compassion fatigue [25] and burnout [26,27] for workers. Individual interventions at the micro level can be very effective with improving one’s own wellbeing; however, individuals can feel constrained by the meso and macro levels of the system.

At the meso level, individuals collaborate within a school or other educational setting that is bound by a common context, vision, and mission. Each school has a unique culture influenced by formal and informal leaders within the building [28,29]. Cultures can be compassionate or toxic [30], fearless, or fearful [31], which influences the wellbeing of individual adults and, by extension, their students.

Schools are organized collectively into a division—the macro level—which is tasked with developing common policies, procedures, and legal directives to organize workplace behaviours and priorities. Given the legislative changes to psychological safety at work combined with widespread teacher and educational worker shortages across Canada [32,33], wellbeing has become a priority at the division level. 

Recently, a greater focus has been taken by researchers [34,35] on the organizational (school) and systemic (division) interventions that are necessary to create safe, caring, and compassionate school cultures for children and youth. One area of specific concern to improve workplace wellbeing in relation to positive mental health and wellbeing could be to reduce stigma and address silence around the distress and stress of the adults in educational settings. 

### 1.3. Silence in Emotional and Mental Health and Wellbeing of Educational Workers

The stigma attached to both diagnosis of and the treatments for emotional and mental health distress is a known challenge to recovery [36,37,38]. While less studied in the field of education [39], stigma is a problem that needs to be addressed.

In education, the ideal worker and “teacher as hero/saviour” myths have long prevailed and is reinforced through popular culture [40] and shows up in one’s professional identity [41], potentially reducing efficacy [42]. The teacher hero/saviour myth is pronounced in television shows and movies that portray the heroic individual who single-handedly turns around difficult and challenging school environments or the demonic teacher or principal who deliberately harms their students and colleagues.

Fictional characters can highlight the impossible standards that educational workers may hold for themselves, their leaders, and colleagues [40,41,42,43], creating the foundation for silence. Safe and caring school cultures require individuals who have the time, supports, and resources to be well [11,37]. Through the investigation of the interventions currently used, this research study aimed to highlight and connect the micro, meso, and macro levels to improve the overall wellbeing of the educational worker workforce.

## 2. Methods

This multi-methods research study used quantitative data collected through an online survey with Likert-style and open-ended questions. As part of a larger study assessing compassion fatigue and burnout in educational workers [1,44], a survey series was constructed to assess compassion fatigue using the ProQOL Version 5 measurement tool [45] and burnout using the Maslach and Jackson burnout inventory [46]. To gather data related to the interventions used by educational workers to manage their stress and distress, a series of checklists and open-ended qualitative questions were included in the survey design.

Survey data were collected at three time points, June 2020, January 2021, and May 2023, via an online survey distributed to potential participants through their membership in the Alberta Teachers Association (ATA), the provincial union for certificated teachers and the ASEBP (Alberta School Employment Plan), the main benefits provider for educational workers, such as administrative assistants, educational assistants, and professional staff. Survey links were made available through internal newsletters, Facebook pages, and Twitter posts. Interviews were held with a sub-sample of volunteer participants who completed the survey and held a variety of job roles between July and September 2020 [44], providing some insights into the different ways that educational workers address their workplace wellbeing.

Institutional ethics were obtained, and survey responses were collected via Survey Alchemer for three weeks at each time point. Unique participants were recruited from across the province of Alberta for each data collection period, although data were not associated with individuals across time, representing a cross-sectional approach. Questions specifically related to the form of intervention used a checklist of common interventions (see Appendix A for complete list), and a series of open-ended questions including “Which of the following supports or resources would you use to feel better?”, an open-ended “other-write-in option”; an open-ended question, “What strategies or activities do you use to feel better (physically, spiritually, emotionally, intellectually, mentally, or socially?”; and “Is there anything else that you would like to tell the research team about your experiences with compassion fatigue, emotional labour, or burnout?”.

## 3. Results

### 3.1. Quantitative Data Analysis

For data analysis, both descriptive and inferential statistics were applied to the aggregate data collected between June 2020 and May 2023 using SPSS Version 28. Descriptive statistics described the interventions used by the participant groups. The entire participant group of all educational workers was divided into two groups for purposes of descriptive analysis. The first group consisted of self-identified teachers and represented the majority of respondents (see Table 1). The second group was labelled as administrators (see Table 2), which included a diverse set of educational workers including principals, educational assistants, school district managers, school secretaries, and transportation workers. This grouping of participants allowed for a more nuanced analysis; however, because of the diversity of work roles represented by the second group of administrators, some caution should be exercised with applying the findings to specific work roles.

In the analysis, descriptive statistics were conducted using SPSS for multiple response questions related to support strategies. Frequencies and percentages were calculated for each strategy across different years of service (0–5, 6–10, 11–15, 16–20, and 21+ years). Cross-tabulations were used to compare the use of support services by year of service, providing insights into the distribution of responses across various strategies.

Prior to data analysis, erroneous entries and missing values were checked, and data were cleaned. Post hoc tests, specifically Bonferroni, explored differences between multiple groups using the number of years of service denoted as 0–5 years, 6–10 years, 11–15 years, 16–20 years, and 21+ years.

### 3.2. Qualitative Data Analysis

Initially, open-ended responses in Year 1 (June 2020) were analyzed using the common thematic codes in response to the question, “What strategies or activities do you use to feel better?”. These codes were then re-used to analyze the data from the Year 2 (January 2021) and Year 3 (May 2023) surveys, using the buckets feature in Survey Alchemer (www.alchemer.com). The buckets feature allows the coder to assign common topics or subjects across multiple surveys and a large dataset (see Table A1). A new code was added when a response was repeated by at least five different survey respondents, resulting in a series of themes that were then categorized as four main interventions aligning with the micro/meso/macro levels: self-directed/individual, professional, community-based, and organizational/system. Because of the large dataset (over 6000 responses across the three data collection periods), frequency counts were used to assign survey responses to codes.

Data were first sorted into codes by the principal investigator and then checked and re-sorted by at least two research assistants for each data collection point, ensuring that the sorting of data into codes was valid. The principal investigator and one lead research assistant were involved in the coding across each data point, ensuring a reliable dataset, while the addition of a new research assistant for each dataset allowed for new perspectives to address problems of bias.

The Supports checklist was developed in discussion with partners from the ATA and ASEBP who have been involved with providing benefits to educational workers for over sixty years. Because this list was not comprehensive, and represented the supports and resources known to be available, the other-write-in option was provided to understand if other supports and resources have been accessed. 

For the other-write-in section of the Supports checklist, the three most common buckets were as follows: No answer (refusal to answer);Accessing mental health professionals in person;Engaging in other activities.

The codes, in random order, used to analyze the responses to Question 23 were as follows:Self-directed strategies;Environmental interventions;Professional intervention strategies;Community-based intervention strategies;Peer or supervisor-directed strategies;Organizational/system-directed strategies;Educational worker strategies;Nothing/no strategies;No answer;Unhealthy intervention/coping strategies.

In response to Question 24, the codes were as follows: No answer (blanks);Importance of work environment/leadership;Assistance, support, and more resources are needed to carry out the job well;Plea for help/lack of support impacting efficacy/ability to carry out the job well;Improve work/classroom conditions;Isolation/no one understands;Description of positive self-care;Survey fatigue.

### 3.3. Findings

The combined analysis of the quantitative and qualitative data collected over the three years suggests alignment with the historical over-emphasis on taking an individual approach to workplace wellbeing as participants were more likely to use self-directed/individual interventions to cope with workplace stress. 

#### 3.3.1. Strategies Used by Teacher Participants for Workplace Wellbeing

Table 1 represents the data collected in response to the checklist of the strategies employed by educators over three years, further broken down by the number of years they have served in their current work role in education.

The results in Table 1 are based on multiple-choice questions from the survey where respondents were allowed to select more than one option for support strategies they used. The percentages shown in the parentheses represent the proportion of respondents in each group (based on years of service) who selected a particular strategy. The percentage is calculated as the number of respondents who selected the strategy (f) divided by the total number of respondents in that group (indicated as N for each year of service) multiplied by 100.

Due to the reduced availability of in-person services in response to the COVID-19 pandemic [47,48], the research team specifically included on the checklist online or virtual therapy appointments rather than the more traditional in-person therapy. As a result, a trend toward a return to in-person therapy is noted in the final year of data collection, with the qualitative responses indicating a preference for meeting with a therapist face to face rather than virtually.

However, despite the availability of supports, the personal support network was the most popular among all groups, with over 87% of respondents across all the years of experience relying on family and friends for support. This reliance continued to be the strongest across the second and third year, suggesting an over-reliance on personal relationships rather than professional, organizational, or systemic supports.

A significant number, representing 70% in each year of service of teachers, turned to a paraprofessional, one-time pain management services such as massage, chiropractic, or physiotherapy. Notably, teachers with 21+ years of experience consulted their family physicians more (50.6%), compared to other groups.

The least popular strategy was the phone help line across all three years and all years of service with usage percentages in all experience categories ranging from 3.7% to 5.2%. Virtual therapy appointments showed an increase in Year 2, especially by respondents with fewer years of experience. This use could have indicated a trend or growing acceptance towards online therapy, but a decrease in Year 3, especially amongst the more experienced educators, may reflect the greater availability of in-person therapy as pandemic restrictions had eased considerably by May 2023 [49].

Additionally, over the three data collection points, a consistent number of participants (40%) indicated use of employer benefits, often noting in their qualitative responses that the amount provided was appreciated but insufficient. Further, a general declining trend in active support networks is a trend in the data, especially among the highly experienced teachers. Given that by the third year, most gyms and other activities coded under active support networks were available, this declining trend may be problematic given the necessity of physical wellbeing to improving overall health and wellness [50].

#### 3.3.2. Differences in Strategy Use Related to Teachers’ Years of Experience

Of note, early career teachers appear more inclined towards modern solutions like online therapy, with later career teachers (21+ years of experience) relying more heavily on family physicians and in-person therapeutic and paraprofessional services. The reason for these differences may be related to availability or trust in different forms of professional intervention; however, the qualitative and quantitative data are silent on specific reasons for these differences.

While teachers across all experience levels heavily rely on personal connections, their choice of additional support systems seems to vary, with newer educators leaning towards modern solutions and veteran educators gravitating towards traditional methods. Over the years, a noticeable shift in preference for the strategies used is evident, providing a place for further research.

#### 3.3.3. Strategies Used by Administrator Participants for Workplace Wellbeing

The second group, labelled as administrators comprised of all other educational worker roles other than Teacher, was also analyzed by years of service (see Table 2). While data offer insight into the strategies employed by administrators, including principals, educational assistants, transportation staff, and managers, specific insights related to work role cannot be made.

The results in Table 2 are based on multiple-choice questions from the survey where respondents were allowed to select more than one option for support strategies they used. The percentages shown in the parentheses represent the proportion of respondents in each group (based on years of service) who selected a particular strategy. The percentage is calculated as the number of respondents who selected the strategy (f) divided by the total number of respondents in that group (indicated as N for each year of service) multiplied by 100.

Similar to the teacher group, the administrator group shows an over-reliance on their personal support network to manage their workplace stress, with high numbers accessing professional services, such as massage and physiotherapy. Online therapy use showed the same increase in Year 2 and then decrease in Year 3, potentially related to COVID-19 pandemic restrictions. As with the teacher group, the administrator group also used the phone help line the least.

Related to years of experience, participants labelled as administrators with 0–5 years of experience start by heavily using online therapy, massages, and personal support networks in Year 1, with a decrease in online therapy in Year 2 but a resurgence in Year 3. Those participants with 6–10 years appear consistent in their use of strategies across the years but with an increasing trend in online therapy and massages. In the 11–15 years group, a decline in the use of online therapy in Year 3 is noted, but personal support and massages remain popular choices. Administrators with 16–20 years appear consistent in their strategies, favouring personal networks and massages, and participants with 21+ years of experience have a marked preference for massages and personal networks, with slightly lesser emphasis on online therapy services.

The reliance on personal support networks to manage workplace stress is significant across all experience groups. Administrators with more experience demonstrated a noticeable trend toward physical wellness resources such as massages and physiotherapy. Online therapy sees varying usage across the years, that may be related to the COVID-19 pandemic restrictions increasing the availability of these services. As with the teacher group, personal support networks and wellness therapies dominated the interventions accessed by administrators.

#### 3.3.4. Overview of Intervention Types Utilized by Participants

The frequency count of the qualitative open-ended questions also demonstrates an over-reliance on individual interventions and personal support networks. Because of the bucket functionality of the Survey Alchemer platform, the data were analyzed in aggregate and not separated by years of service or the teacher/administrator grouping. Further, some responses included examples of several different strategies used, so the total number reflects the number of responses that included the specific intervention type, not the total number of responses from participants. 

Table 3 reveals the general forms of intervention most commonly used by participants across the three data collection points. Self-directed strategies included individual wellness practices, such as going to the gym, speaking with a friend, or praying. Professional interventions involved accessing expert guidance from trained professionals, and responses included meeting with a family doctor, accessing a physiotherapist, or attending psychologist appointments. Community-based interventions were non-workplace social spaces such as churches, synagogues, temples, mosques, and book clubs. Leadership or colleague interventions were related to mentorship and workplace supports, such as discussing workload with a caring school principal or meeting socially with other members of one’s department. Organizational and system directed interventions clustered governmental, union, and school district policies and procedures related to the workplace.

Respondents made a clear distinction between online and in-person therapy with a noted preference for in-person, with 107 respondents in the Year 3 data specifically adding in-person therapy in the open-ended option as a desired intervention.

In the analysis of the data collected in Year 3, a notably different self-directed intervention, coded as environmental interventions, emerged. These responses specifically mentioned incorporating land-based or outdoor activities into one’s repertoire of coping strategies or getting outside and being in nature to restore one’s mental health after a stressful event. While this intervention aligns with a self-directed strategy, the data coders noted the specificity of reference to the calming effects of going outside was a different than the largely indoor or online self-directed strategies survey participants had highlighted previously. This new theme suggests a stronger desire for outdoor professional learning and time for educational workers to relieve their symptoms of compassion fatigue and burnout.

Lastly, in reference to the final questions, “is there anything else you would like to tell the research team”, across the three years of data collection, the two most referenced interventions needed were related to the school and system conditions impacting the respondents’ ability to do their work well. While individual interventions were mentioned by a very small number of respondents, the majority of the most meaningful and impactful changes to improve their wellbeing were related to improved working conditions, adequate resources to improve student learning, and the importance of a work culture and leadership team that prioritized wellbeing (see Table 4).

Collectively, the qualitative and quantitative data support the development of wellbeing initiatives to explicitly address workplace culture, workload, and leadership rather than focusing on individual or self-directed interventions.

## 4. Discussion

The workplace psychological hazards of compassion fatigue and burnout are eroding the wellbeing of teachers, school leaders, and the support staff working in schools and other educational settings across Alberta [1]. Expanding the accessible interventions are sorely needed to ensure that the adults who care for the children and youth in schools can thrive.

As highlighted by the data analysis, a very high proportion of the survey respondents have self-directed strategies and strong personal support networks to help them in times of stress, but these individual interventions are often not sufficient to relieve their distress. Of highest importance, time and space are required during the workday for educational workers to decompress and deal with symptoms of stress. Educational workers are asking for organizational and system supports to improve their workplace wellbeing.

As noted by one respondent in 2023, 


*The most difficult part is that my principal and system know that this is an issue that is facing so many educators, but they do nothing about it. There has been an acknowledgment of this but nothing to help ameliorate it. There has been no decrease in workload, no extra resources. There is a prevailing feeling like I am responsible for my own self-care, and if I am feeling this way it is my own fault.*
(Survey open-ended response)

Adding to the systemic stresses is the guilt felt by educational workers with regards to taking time away from their schools and classrooms to adequately address their mental and emotional wellbeing. 


*We feel guilty about taking a mental health day. There’s so much talk about taking care of ourselves but there aren’t subs to take our jobs. Also, our benefits don’t cover much for therapy sessions. The last time I used [Benefits Program], it was an ok experience, but it’s hard starting over with a new therapist every time. It’s also time and effort to work with one even if they aren’t the best fit.*
(Survey open-ended response)

Educational workers will often refuse to take time away from teaching or the school to address distress in the summer break because of the additional work required and feelings of guilt for using their contracted and negotiated days of personal business leave. Addressing the lack of resources allocated to funding public education needs to be a foundational aspect of any workplace wellbeing program [51,52]. No amount of extra intervention will aid in the recovery of educational workers if they do not feel empowered to take proactive steps to meet with health professionals when assistance is required.

Finally, a mixture of positive and maladaptive coping strategies was described in the open-ended responses. Several respondents referenced using alcohol, marijuana, and other addictive substances to cope with their workplace distress. While not a focus of the analysis for this study, the negative impact of maladaptive coping strategies on educational worker functioning is an area for further study. 

### Limitations

The participants were recruited online via Facebook and Twitter, and while demographic questions were specific to Alberta, the actual location of the respondents is not known. Although the data were collected at three time points, because of the anonymous online nature of the survey, the research team does not know if the same people responded to different collection points. Lastly, data were collected during the three-year height of the COVID-19 pandemic, so the type of intervention used was likely influenced by the restrictions of the time. Further research is needed to understand if the type of intervention selected by respondents changed in response to the lifting of all restrictions in the post-pandemic period.

## 5. Conclusions

High levels of burnout and secondary traumatic stress across the education field have been noted [1], stressing the importance of developing workplace wellbeing interventions to ensure a healthy workforce. However, this current study suggests that programs or policies focused primarily on self-directed or individual strategies are not well received by educational workers because they already know the individual efforts required to safeguard and protect their overall wellbeing. As described by a respondent,


*This is a job, and it should not erode my mental and physical health in the ways that it has. I have never been diagnosed with mental health illnesses before, and it’s the working conditions that made me sick. I am 15 years in and mentally ill- because of my JOB…I will complete every survey, but at this point, it feels like I am screaming into a void. I sincerely hope teaching doesn’t kill me.*
(Survey open-ended response)

Integrating systemic and organizational interventions that address the core conditions of class size and complexity, reporting and assessment requirements, and adequate hiring of staff are necessary to complement the individual efforts of educational workers to feel well. 

Further, specific educational interventions that positively influence the wellbeing for students, and the staff who work with them, related to teaching and learning need to be constructed, prioritized, and shared. Respondents suggested that strategies such as daily preparation time for lesson planning and assessment, creating fair policies for promotion and supervision, aligning competence with teaching load and course assignments, constructing adequate classroom space for children and youth, and training for educational assistants in special education and early literacy intervention would have a positive impact on their overall wellbeing. 

Building rest, support, and recovery into the school culture supported through district policy would have the largest impact on individuals’ ability to cope with stressful situations. Given the data collected over this three-year study, educational workers have clearly identified that they do not need programs to build stronger individual habits unless these programs include time and resources to meet their wellness goals during their workday. They want to take responsibility for their own wellbeing, but they require the working conditions to do so.

Other research suggests that compassionate and caring school cultures are critical for student flourishing [53,54,55] and may have a reciprocal positive impact on the adults who work with them [52]. Developing institutional and organizational policies that support a positive school culture is foundational for reducing employee attrition and increasing retention. School and governmental authorities need to prioritize healthy workplaces first because educational employees’ workplaces are the environments within which students learn. 

## Figures and Tables

**Table 1 healthcare-12-02200-t001:** Strategies teachers used by years of service.

Years	Strategies	How Many Years of Service Do You Have in Your Current Work Role in Education?				
		0–5 Years	6–10 Years	11–15 Years	16–20 Years	21+ Years
		(N = 295)	(N = 330)	(N = 280)	(N = 252)	(N = 325)
Year 1 (June 2020)	Online therapy services	85 (28.8%)	96 (29.1%)	64 (22.9%)	59 (23.4%)	67 (20.6%)
	Phone help line	11 (3.7%%)	13 (3.9%)	11 (3.9%)	12 (4.8%)	17(5.2%)
	Personal support network (friends, family)	273 (92.5%)	298 (90.3%)	244 (87.1%)	228 (90.5%)	283 (87.1%)
	Active support network (gym, run club, yoga class)	208 (70.5%)	217 (65.8%)	179 (63.9%)	147 (58.3%)	184 (56.6%)
	Employer benefits or assistance program	109 (36.9%)	119 (36.1%)	99 (35.4%)	85 (33.7%)	104 (32.0%)
	Family physician	85 (28.8%)	104 (31.5%)	112 (40.0%)	112 (44.4%)	142 (43.7%)
	Massage, chiropractic services, or physiotherapy services	227 (76.9%)	253 (76.7%)	210 (75.0%)	185 (73.4%)	245 (75.4%)
Year 2 (January 2021)		**0–5 years**	**6–10 years**	**11–15 years**	**16–20 years**	**21+ years**
		**(N = 316)**	**(N = 341)**	**(N = 296)**	**(N = 249)**	**(N = 336)**
	Online therapy services	123 (38.9%)	131 (38.4%)	89 (30.1%)	54 (21.7%)	77 (22.9%)
	Phone help line	7 (2.2%)	21 (6.2%)	13 (4.4%)	4 (1.6%)	15 (4.5%)
	Personal support network (friends, family)	266 (84.2%)	310 (90.9%)	257 (86.8%)	212 (85.1%)	306 (91.1%)
	Active support network (gym, run club, yoga class)	215 (68.0%)	242 (71.0%)	179 (60.5%)	153 (61.4%)	196 (58.3%)
	Employer benefits or assistance program	115 (36.4%)	138 (40.5%)	102 (34.5%)	84 (33.7%)	114 (33.9%)
	Family physician	100 (31.6%)	129 (37.8%)	127 (42.9%)	107 (43.0%)	170 (50.6%)
	Massage, chiropractic services, or physiotherapy services	241 (76.3%)	271 (79.5%)	221 (74.7%)	191 (76.7%)	244 (72.6%)
Year 3 (May 2023)		**0–5 years**	**6–10 years**	**11–15 years**	**16–20 years**	**21+ years**
		**(N = 112)**	**(N = 253)**	**(N = 222)**	**(N = 252)**	**(N = 327)**
	Online therapy services	39 (34.8%)	83 (32.8%)	59 (26.6%)	58 (23.0%)	71 (21.7%)
	Phone help line	5 (4.5%)	10 (4.0%)	10 (4.5%)	8 (3.2%)	9 (2.8%)
	Personal support network (friends, family)	99 (88.4%)	226 (89.3%)	190 (85.6%)	223 (88.5%)	277 (84.7%)
	Active support network (gym, run club, yoga class)	89 (79.5%)	170 (67.2%)	130 (58.6%)	142 (56.3%)	174 (53.2%)
	Employer benefits or assistance program	51 (45.5%)	120 (47.4%)	75 (33.8%)	103 (40.9%)	129 (39.4%)
	Family physician	49 (43.8%)	125 (49.4%)	107 (48.2%)	125 (49.6%)	179 (54.7%)
	Massage, chiropractic services, or physiotherapy services	93 (83.0%)	210 (83.0%)	165 (74.3%)	196 (77.8%)	242 (74.0%)

**Table 2 healthcare-12-02200-t002:** Strategies administrators used by years of experience.

Years	Strategies	How Many Years of Service Do You Have in Your Current Work Role in Education?				
		0–5 Years	6–10 Years	11–15 Years	16–20 Years	21+ Years
		(N = 135)	(N = 90)	(N = 89)	(N = 79)	(N = 129)
Year 1, June 2022	Online therapy services	37(27.4%)	21 (23.3%)	23 (25.8%)	22 (27.8%)	20 (15.5%)
	Phone help line	6 (4.4%)	2 (2.2%)	4 (4.5%)	5 (6.3%)	8 (6.2%%)
	Personal support network (friends, family)	118 (87.4%)	77 (85.6%)	74 (83.1%)	69 (87.3%)	108 (83.7%)
	Active support network (gym, run club, yoga class)	92 (68.1%)	60 (66.7%)	46 (51.7%)	46 (58.2%)	77 (59.7%)
	Employer benefits or assistance program	58 (43.0%)	30 (33.3%)	30 (33.7%)	28 (35.4%)	42 (32.6%)
	Family physician	47 (34.8%)	34 (37.8%)	34 (38.2%)	27 (34.2%)	57 (44.2%)
	Massage, chiropractic services, or physiotherapy services	103 (76.3%)	72 (80.0%)	70 (78.7%)	56 (70.9%)	103 (79.8%)
Year 2, January 2021		**0–5 years**	**6–10 years**	**11–15 years**	**16–20 years**	**21+ years**
		**(N = 45)**	**(N = 59)**	**(N = 65)**	**(N = 40)**	**(N = 94)**
	Online therapy services	9 (20.0%)	22 (37.3%)	20 (30.8%)	9 (22.5%)	17 (18.1%)
	Phone help line	2 (4.4%)	5 (8.5%)	2 (3.1%)	1 (2.5%)	5 (5.3%)
	Personal support network (friends, family)	39 (86.7%)	52 (88.1%)	57 (87.7%)	35 (87.5%)	83 (88.3%)
	Active support network (gym, run club, yoga class)	30 (66.7%)	40 (67.8%)	46 (70.8%)	18 (45.0%)	55 (58.5%)
	Employer benefits or assistance program	19 (42.2%)	28 (47.5%)	26 (40.0%)	13 (32.5%)	36 (38.3%)
	Family physician	17 (37.8%)	28 (47.5%)	30 (46.2%)	14 (35.0%)	46 (48.9%)
	Massage, chiropractic services, or physiotherapy services	37 (82.2%)	47 (79.7%)	55 (84.6%)	26 (65.0%)	70 (74.5%)
Year 3, May 2023		**0–5 years**	**6–10 years**	**11–15 years**	**16–20 years**	**21+ years**
		**(N = 100)**	**(N = 88)**	**(N = 76)**	**(N = 70)**	**(N = 128)**
	Online therapy services	32 (32.0%)	28 (31.8%)	17 (22.4%)	23 (32.9%)	29 (22.7%)
	Phone help line	6 (6.0%)	4 (4.5%)	5 (6.6%)	3 (4.3%)	3 (2.3%)
	Personal support network (friends, family)	85 (85.0%)	75 (85.2%)	66 (86.8%)	61 (87.1%)	111 (86.7%)
	Active support network (gym, run club, yoga class)	59 (59.0%)	52 (59.1%)	39 (51.3%)	36 (51.4%)	81 (63.3%)
	Employer benefits or assistance program	48 (48.0%)	45 (51.1%)	22 (28.9%)	25 (35.7%)	46 (35.9%)
	Family physician	40 (40.0%)	41 (46.6%)	34 (44.7%)	31 (44.3%)	60 (46.9%)
	Massage, chiropractic services, or physiotherapy services	74 (74.0%)	65 (73.9%)	56 (73.7%)	53 (75.7%)	100 (78.1%)

**Table 3 healthcare-12-02200-t003:** Responses to “What strategies or activities do you use to feel better (physically, spiritually, emotionally, intellectually, mentally, or socially)?”.

Year of Data Collection	Form of Intervention	Percentage of Responses
Year 1 (June 2020)	Self-directed strategies	98.4%, n = 1807
	Professional interventions	10.8%, n = 198
	Community-based interventions	9.3%, n = 198
	Leadership or colleague interventions	7.9%, n = 146
	Organizational/system directed interventions	11.7%, n = 215
Year 2 (January 2021)	Self-directed strategies	97.8%, n = 987
	Professional interventions	22.3%, n = 225
	Community-based interventions	14.1%, n = 142
	Leadership or colleague interventions	9.4%, n = 95
	Organizational/system directed interventions	3.2%, n = 32
Year 3 (May 2023)	Self-directed strategies	84.5%, n = 983
	Environmental intervention	22.4%, n = 261
	Professional interventions	21.3%, n = 248
	Leadership or colleague interventions	3.4%, n = 40
	Organizational/system directed interventions	0.9%, n = 11
	Educational worker interventions	2.1%, n = 25
	None/nothing	2.5%, n = 29
	No answer (blanks)	8.7%, n = 101
	Unhealthy interventions (maladaptive coping)	2.1%, n = 25

**Table 4 healthcare-12-02200-t004:** Coded responses to “Is there anything else that you want to tell the research team about your experiences with compassion fatigue, emotional labour, or burnout?”.

Year of Data Collection	Form of Intervention	Percentage of Respondents
Year 1 June 2020 N = 1088	Improve work/classroom conditions	40.6%, n = 442
	Plea for Help/Lack of support impacting efficacy/ability to do job effectively	32.7%, n = 356
	Importance of work environment/leadership	20.8%, n = 226
	Societal judgement (parents, government, general community) impacting efficacy and ability to be effective	19.2%, n = 209
	Importance of positive self-care	11.9%, n = 129
Year 2 January 2021 N = 661	Plea for help/lack of support impacting efficacy/ability to do job effectively	60.7%, n = 371
	Improve work/classroom conditions	47.3%, n = 289
	Societal judgement (parents, government, general community) impacting efficacy and ability to be effective	35.7%, n = 218
	Call to action/do something about burnout and compassion fatigue	14.9%, n = 91
	Importance of work environment/leadership	11.1%, n = 68
Year 3 May 2023 N = 1072	No answer	40.3%, n = 432
	Importance of work environment/leadership	23.5%, n = 252
	Assistance, support, and more resources needed to do job well	21.2%, n = 227
	Plea for help/lack of support impacting efficacy/ability to do job effectively	17.4%, n = 186
	Improve work/classroom conditions	13.3%, n = 143

## Data Availability

Data are contained within the article.

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
