# Peer review of "A Longitudinal Multi-Method Inquiry of Educational Workers’ Use of Interventions for Positive Mental Wellbeing"

_healthcare, 2024, doi:10.3390/healthcare12222200_

Round 1

Reviewer 1 Report

Comments and Suggestions for Authors

Recommendations for Authors:

  1. The introduction provides a good overview of workplace wellbeing in the Canadian context and the theoretical framework. However, it could be strengthened by:
  • Including more recent references on burnout and compassion fatigue specifically in educational settings
  • Providing a clearer rationale for why this study is needed and how it fills gaps in the current literature
  • Explaining the choice of Alberta as the study location and any unique contextual factors
  1. More details could be provided on:
  • Survey development and validation process
  • Specific qualitative analysis methods used
  • Rationale for grouping participants into "teachers" and "administrators" categories
  • Any steps taken to ensure data quality and reliability across the 3 years
  1. The results section would benefit from:
  • Clearer organization and subheadings to guide the reader
  • More interpretation alongside the descriptive statistics
  • Clearer explanation of which results address which research questions
  • Consideration of statistical significance for quantitative comparisons
  • More substantive quotes from qualitative data to illustrate key themes

Additional comments for authors:

  • The abstract could be more focused and concise
  • Consider reorganizing results/discussion to more clearly address research questions
  • Provide more context on how COVID-19 may have impacted findings
  • Discuss limitations more thoroughly, including potential sampling biases
  • Strengthen connections to existing literature in the discussion
  • Consider implications for policy and practice more explicitly

Author Response

Thank you for your comments.

  1. The introduction provides a good overview of workplace wellbeing in the Canadian context and the theoretical framework. However, it could be strengthened by:
  • Including more recent references on burnout and compassion fatigue specifically in educational settings

We added some more recent references, although most of our references are already within the last five years.

  • Providing a clearer rationale for why this study is needed and how it fills gaps in the current literature

The rationale has been made clearer, but is already stated in the first paragraph (high levels of compassion fatigue and burnout due to the pandemic, frequent weather disasters because of climate change, and a shortage of trained educators to lessen the workload.

  • Explaining the choice of Alberta as the study location and any unique contextual factors

Alberta was selected because the study was locally developed and funded. Information about the Alberta context was added in the revised manuscript.

  1. More details could be provided on:
  • Survey development and validation process

These details have been added.

  • Specific qualitative analysis methods used

Information was revised to more clearly reflect the form of qualitative analysis used.

  • Rationale for grouping participants into "teachers" and "administrators" categories

This rationale, as well as who was considered members of each group, was already given within the paper.

  • Any steps taken to ensure data quality and reliability across the 3 years

Clarity provided within revised manuscript.

  1. The results section would benefit from: 
  • Clearer organization and subheadings to guide the reader

The manuscript already had headings, but tried to make them more visible within the journal guidelines.

  • More interpretation alongside the descriptive statistics

Some additional intepretation was provided however, we needed to stay within the character limit of the journal.

  • Clearer explanation of which results address which research questions

The paper only had one research question, so all the results address it.

  • Consideration of statistical significance for quantitative comparisons

Significance was detailed within the tables.

  • More substantive quotes from qualitative data to illustrate key themes

Addressed as much as possible to stay within character limit.

Additional comments for authors:

  • The abstract could be more focused and concise

The abstract was reduced as much as possible to comply with journal standards.

  • Consider reorganizing results/discussion to more clearly address research questions

Given the manuscript only addressed on research question, the results and discussion section were revised only for clarity.

  • Provide more context on how COVID-19 may have impacted findings

The influence of COVID was described in each section as well as the limitations section.

  • Discuss limitations more thoroughly, including potential sampling biases

We discussed the limitations as much as possible given the manuscript length.

  • Strengthen connections to existing literature in the discussion

Some additions were made to link the literature to the discussion. Very little has been done on the interventions used by educational workers to improve their wellbeing.

  • Consider implications for policy and practice more explicitly

Policy implications are beyond the scope of this paper. Practice is addressed throughout the manuscript.

Reviewer 2 Report

Comments and Suggestions for Authors

Thank you for the opportunity to review the article “A Longitudinal Multi-Method Inquiry of Educational Workers Use of Interventions for Positive Mental Wellbeing

The article deals with a very relevant topic and is well founded with relevant scientific evidence. The methods adopted are appropriate and well founded.

The results are presented primarily in a descriptive way and could perhaps provide more evidence of the analysis of inferences.

The discussion of the results and the conclusion responds to the objectives of the article. The results of the research results that may have important impacts on the mental health of teachers in an international audience

Author Response

Thank you for the opportunity to review the article “A Longitudinal Multi-Method Inquiry of Educational Workers Use of Interventions for Positive Mental Wellbeing

The article deals with a very relevant topic and is well founded with relevant scientific evidence. The methods adopted are appropriate and well founded.

The results are presented primarily in a descriptive way and could perhaps provide more evidence of the analysis of inferences.

The discussion of the results and the conclusion responds to the objectives of the article. The results of the research results that may have important impacts on the mental health of teachers in an international audience

Thank you for your positive comments and feedback.

Reviewer 3 Report

Comments and Suggestions for Authors

Dear Authors

I was delighted to receive the invitation to review your manuscript. 

This is a very important topic, and the COVID-19 pandemic has given it even more importance.

However, I have a few comments to make:

The tables could be better organized if you put the percentage between relatives next to the absolute frequency. As it is, they are very long and confusing. They should read: 37 (27.4%). Halve the number of lines in the tables.

In the quantitative data analysis, they talk about the One Way ANOVA test, but I've looked and I don't see any results of this test. Either it's your fault or you didn't carry out the test.

You should deepen the discussion, as it is very short.

You should also include the practical implications and the theoretical implications.

My Best Regards

Author Response

Thank you for your comments.

The tables could be better organized if you put the percentage between relatives next to the absolute frequency. As it is, they are very long and confusing. They should read: 37 (27.4%). Halve the number of lines in the tables.

The tables have been updated.

In the quantitative data analysis, they talk about the One Way ANOVA test, but I've looked and I don't see any results of this test. Either it's your fault or you didn't carry out the test.

We removed the reference to ANOVA with the reason:

We are delete this section from the work since ANOVA analysis was not conducted for the dataset. The dataset used for this paper is categorical in nature, specifically, service years in various educational roles across different strategies for the years 2021, 2022, and 2023. ANOVA can not be used because, ANOVA (Analysis of Variance) is typically used for analyzing differences in means across different groups when the data is continuous. Since your data is categorical, consisting of counts and percentages, ANOVA would not be appropriate for this dataset.

You should deepen the discussion, as it is very short.

We had a short discussion to stay within the character count requirement. The key findings are woven throughout.

You should also include the practical implications and the theoretical implications.

Included as much as possible.

Round 2

Reviewer 3 Report

Comments and Suggestions for Authors

Dear authors

Thank you for making the changes I proposed.

My Best Regards